# Interpretability and Generalization Bounds for Learning Spatial Physics

**Alejandro F. Queiruga** [1]   **Theo Gutman-Solo** [2]   **Shuai Jiang** [3]

## Abstract

While there are many applications of machine learning (ML) to scientific problems that *look* promising during training, achieving low training error does not guarantee convergence to the correct physics or generalization beyond the span of the training set. Using numerical analysis techniques, we rigorously quantify the accuracy, convergence rates, and generalization bounds of certain ML models applied to linear differential equations (DEs) for parameter discovery or forward problem solving. Beyond the quantity and discretization of data, we identify that the function space of the data is critical to the generalization of the model. A similar lack of generalization is empirically demonstrated for commonly used models, including physics-specific techniques. Counterintuitively, we find that different classes of models can exhibit opposing generalization behaviors. Based on our theoretical analysis, we also introduce a new mechanistic interpretability lens on scientific models whereby Green's function representations can be extracted from the weights of black-box models. Our results inform a new cross-validation technique for measuring generalization in physical systems, which can serve as a benchmark.

## 1. Introduction

The development of robust and rigorous a priori estimates for numerical methods is a major victory for scientific computing in the past half-century. These estimates give geometric flexibility and convergence guarantees, laying the groundwork for complex physics simulations across a wide range of applications. The basis of the theory begins with the humble 1D Poisson equation, studied by virtually all numerical analysts and serving as the bedrock for computational physics and engineering.

The canonical Poisson equation on the unit interval with homogeneous Dirichlet boundary conditions is given by

$$-k\frac{d^2u}{dx^2} = f(x), \qquad u(0) = u(1) = 0. \qquad (1)$$

where $u(x), f(x), k$ are the solution, forcing function, and (constant) material coefficient, respectively. The solution to Eq. (1) can be obtained using the Green's function which maps a function $f(x)$ to the solution $u(x)$ as follows:

$$u(x) = \int_0^1 f(s)G(s,x)\,\mathrm{d}s + (u_1 - u_0)x + u_0 \qquad (2)$$

where

$$G(s,x) = \begin{cases} (1-s)x & \text{if} \quad x < s \\ (1-x)s & \text{if} \quad x \ge s \end{cases}. \qquad (3)$$

In the ML context, the governing DE is typically unknown, and we are given discrete evaluations of forcing functions $f(x_i)$ and solutions $u(x_i)$ with $x_i \in (0,1)$ corresponding to a sensor grid or pixels from a camera. We then seek to train a model approximating the solution operator that can compute an unknown $u$ given an unseen $f$. While the DE is continuous, the act of measurement discretizes all objects. For simplicity, we only consider uniformly spaced points $x_i = i\Delta x$ for $i = 0, \ldots, N_{\text{grid}}$.

The simplest black-box model is a linear operator given by

$$\boldsymbol{u} = \boldsymbol{W}\boldsymbol{f} \qquad (4)$$

where $\boldsymbol{W} \in \mathbb{R}^{N_{\text{grid}} \times N_{\text{grid}}}$ and is trained via the usual gradient descent on mean squared error (MSE) loss, $\mathcal{L} = \sum \|\boldsymbol{u} - \boldsymbol{W}\boldsymbol{f}\|_2^2$. A natural question is what matrix this model learns: would the matrix $\boldsymbol{W}$ correspond to matrices derived using Green's function or is it less structured?

We show the specific matrices in Fig. 1. We first construct a dataset using analytical solutions with cubic forcing functions, and achieve low train and test error on that space resulting in a learned matrix $\boldsymbol{W}_{p=3}$. The learned matrix (second row) qualitatively resembles the analytically derived matrix from Green's function (first row), but is quite far from the "truth" when we look at the inverse $\boldsymbol{W}_{p=3}^{-1}$. In

[1] OpenAI, San Francisco, CA, USA [2] Google, Mountain View, CA, USA [3] Sandia National Laboratories, Albuquerque, NM, USA. Correspondence to: Alejandro F. Queiruga <Alejandro.Queiruga@gmail.com>.

*Proceedings of the 43rd International Conference on Machine Learning*, Seoul, South Korea. PMLR 306, 2026. Copyright 2026 by the author(s).

general, the model will appear uninterpretable and fail to generalize to out-of-distribution (OOD) datasets. However, with careful construction of the dataset (bottom of Fig. 1), we can recover both a full Green's function discretization, $\boldsymbol{A}$, and a discretization of the differential operator, $\boldsymbol{L} = \boldsymbol{A}^{-1}$.

To formalize, let $W$ denote the learned model parameters and $\mathcal{L}(W, \mathcal{F})$ denote the loss, MSE in our case, evaluated on solution-forcing pairs $u, f$ drawn from function space $\mathcal{F}$ (defined in Sec. 3.1). We define a model as *generalizable* if, after minimizing $\mathcal{L}(W, \mathcal{F}_{\text{train}})$, it achieves comparably low error on a distinct function space $\mathcal{F}_{\text{test}} \neq \mathcal{F}_{\text{train}}$:

$$\mathcal{L}(W, \mathcal{F}_{\text{test}}) \lesssim \mathcal{L}(W, \mathcal{F}_{\text{train}}). \qquad (5)$$

The symbol $\lesssim$ denotes a small tolerance. We empirically observe that the OOD error can be orders of magnitude larger than the training error, $\mathcal{L}_{\text{test}} \gg 10^8 \times \mathcal{L}_{\text{train}}$, making the OOD transition stark and unambiguous.

To assist in understanding whether a model is generalizing or not beyond just evaluating losses, we introduce a mechanistic interpretation technique akin to viewing attention maps in DINO (Caron et al., 2021) utilizing Green's function. Rather than use the analytical Green's function as a tool to model or solve PDEs, we use it to visually understand if a certain model has the capability to generalize beyond the training set or not.

We also design a new cross-validation procedure that trains a given ML model with a dataset on a subspace, and then evaluates the error on datasets generated on different *subspaces*. The datasets are varied not only by the grid spacing $\Delta x$, but also by the function class $f \sim \mathcal{F}(type, p)$. We consider five classes of models spanning from underparameterized physics-informed fits to overparameterized black-box neural networks, and present the following contributions:

1. *Underparameterized models with an inductive bias of known physical equations* (Thm. 3.1, Sec. 3.2, Fig. 4a) — We demonstrate both empirically and theoretically that the estimated parameter $w$ converges to the true parameter $k$ at a rate that depends on the grid spacing $\Delta x$ and finite difference order $q$, and that *increases* with the order of the training function basis $p$.

2. *Exact parametrization for linear problems* (Thm. 3.2, Sec. 3.3, Fig. 4b) — We derive an exact form for the discrete solution operator $\boldsymbol{A}$ and theoretically and empirically demonstrate that the learned weights only converge to a subspace of $\boldsymbol{A}$ determined by the basis of the training data. The grid size $\Delta x$ does not change the error, but changes the scaling of learned parameters.

3. *Overparameterized neural models* (Sec. 4.2, Figs. 4c–4d) — We empirically demonstrate that deep models *are not guaranteed to generalize* outside of the training data, and that subspace generalization is also not guar-

anteed. We introduce the Green's function mechanistic interpretation visualization in this context.

4. *Methods designed for DE learning* (Sec. 4.2, Fig. 5) — We observe that the DeepONet and Fourier Neural Operator also do not always generalize, similarly to black-box models.

5. *Physics informed losses* (Sec. 4.2, Fig. 6) — We demonstrate that PINNs and Physics Informed DeepONets, which incorporate prior DE knowledge, exhibit similar failure modes.

Surprisingly, we find that the data requirements and generalization behavior for each model class can vary dramatically, and in some cases, are even contrary to one another. For example, increasing the polynomial degree of the training data reduces error for the linear model but increases error for the finite-difference parameter fit.

## 2. Background

Applications of ML in science span across "white-box" to "black-box". White-box methods, such as SINDy and symbolic regression, allow for the user to obtain closed-form, analytical formulas (Rudy et al., 2017; Udrescu & Tegmark, 2020). Black-box methods such as Neural Operators or DeepONets generally use NNs to learn dynamics (Raissi et al., 2019; Lu et al., 2021a), exchanging interpretability for greater flexibility. Between these two approaches, techniques have been developed that respect physical constraints such as conservation laws or other fundamental identities (Hansen et al., 2023; Patel et al., 2022; Trask et al., 2022). However, the adoption of these methods in high-consequence scenarios is lacking due to not understanding the exact failure modes.

To this end, recent studies have demonstrated that Neural ODEs and residual connections can overfit to time series data and may not learn true continuous models (Queiruga et al., 2020; Ott et al., 2021; Sander et al., 2022), whereas higher order time discretization can elicit generalization (Zhu et al., 2022; Krishnapriyan et al., 2023). For spatiotemporal models, Krishnapriyan et al. (Krishnapriyan et al., 2021) demonstrated that PINNs will fail without controlled training strategies, whereas Sakarvadia et al. (Sakarvadia et al., 2025) showed that PINOs fail to generalize across spatial resolutions. Other recent work addresses resolution-related challenges directly: ReNO (Bartolucci et al., 2023) and RINO (Bahmani et al., 2025) introduce aliasing-free and discretization-invariant operator architectures that can be evaluated at arbitrary resolutions. These advances are orthogonal to our consideration: we study generalization when the test *function space* differs from the training set, not when the discretization differs.

Our work draws on two complementary lines of prior re-

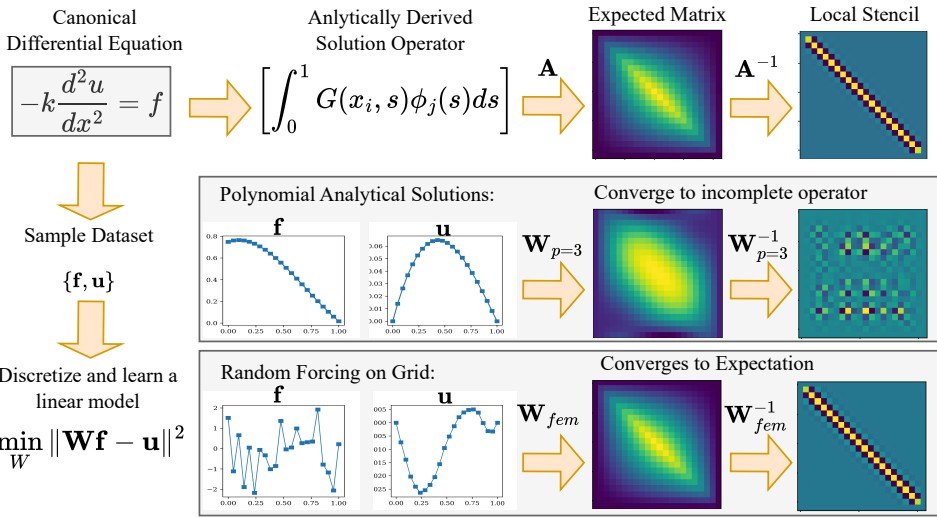

*Figure 1.* Expectations when learning a black box linear model, $\boldsymbol{u} = \boldsymbol{W}\boldsymbol{f}$. Despite some visual similarity to the Green's function operator (top), naive approaches are not guaranteed to converge to the true operator. The issue lies in the sampling procedure of the training data. The naive approach of using polynomial analytical solutions (illustrated as cubic polynomials) can achieve machine precision MSE loss, but even with infinitely many examples, will not converge to the general solution (center). Altering the construction of the training data, i.e. by using random piecewise linear functions (bottom) can recover the true operator. This allows for the extraction of a familiar discrete differential equation operator (right).

search. Green's functions have been used in ML to construct solution operators and to prove data-requirement bounds for elliptic equations (Gin et al., 2021; Boullé et al., 2022; 2023). Separately, mechanistic interpretability has shown that known algorithms can be recovered by directly inspecting the weights of trained models (Nanda et al., 2023); within SciML, related work has drawn analogies between transformer architectures and quadrature, nonlocal operators, or kernel functions (Cao, 2021; Nguyen et al., 2022; Yu et al., 2024). We bridge these threads by treating the Green's function as both a theoretical anchor for generalization bounds and an interpretability lens on learned weights.

## 3. Theoretical Results

### 3.1. Dataset Construction

To explore how the function space of the training data impacts the model, we construct different datasets by sampling from disparate function spaces. Let $\mathcal{F}(type, p)$ denote a function space of a type (e.g., polynomial, FEM, cosine, sine, etc.) with $p$ terms. Given the function class, the dataset is constructed by sampling functions evaluated on a grid with points $x_i \in i\Delta x$ with $\Delta x = 1/N_{\text{grid}}$ (e.g., the spacing of the sensor grid). Entries in the dataset are $\{x_i, f(x_i), u(x_i)\}$ from functions sampled from $f \sim \mathcal{F}(type, p)$ and $u$ obtained from analytical differentiation. In experiments, we consider function types of polynomial, sine $\sin(k\pi x)$, cosine $\cos(k\pi x)$, and piecewise linear (e.g. FEM).

### 3.2. Parameter Learning With a Discretized DE

Consider the case where inductive bias of the DE is known, through either prior knowledge or a system identification method, but its physical parameter is unknown. We fit the discretized Poisson equation for a parameter $w$, corresponding to the true value is $k$ from Eq. 1, such that $-w\frac{d^2u(x_i)}{dx^2} \approx f(x_i)$ over the data. The DE is approximated using a standard finite difference stencil, such as the three-point stencil (FD-2),

$$\frac{d^2u}{dx^2} = \frac{u_{i-1} - 2u_i + u_{i+1}}{\Delta x^2} + \mathcal{O}(\Delta x^2). \qquad (6)$$

We can estimate $w$ given the data $\left\{u^{(i)}, f^{(i)}\right\}_{i=1}^{N}$ by analytically minimizing the MSE loss over the $N$ examples

$$\frac{1}{N}\sum_{n=0}^{N}\sum_{i=1}^{N_{grid}-1}\left(\frac{w}{\Delta x^2}\left(u_{i-1}^{(n)} - 2u_i^{(n)} + u_{i+1}^{(n)}\right) + f_i^{(n)}\right)^2. \qquad (7)$$

The following a priori estimate shows that the relative error $e = |w-k|/|k|$ can be bounded from above for data sampled from polynomials:

**Theorem 3.1.** *Learning the parameter $k$ using a finite difference stencil of order $q$ given polynomial training data of degree $p$ on a grid of spacing $\Delta x$ results in $w = k$ when $p < q$, and an error*

$$\frac{|w-k|}{|k|} = \mu_q\Delta x^q + \sum_{m=q+1}^{p}\mu_m\Delta x^m \approx \mu_q\Delta x^q \qquad (8)$$

*when $p \geq q$, for constants $\mu_m$ depending on the truncation error coefficients of the finite difference stencil.*

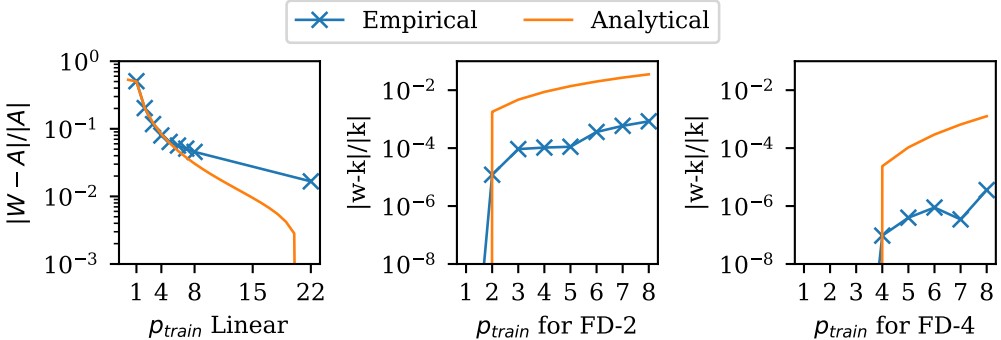

*Figure 2.* Comparison of numerical experiments to analytically derived ML parameters from Theorems 3.1 and 3.2 trained on polynomial spaces with $N_{grid} = 22$. (Left) The observed error for the linear matrix matches predictions for lower order $p$. (Middle and right) For the finite difference model, the analytical assumptions overestimate the error, but the trend matches with observations.

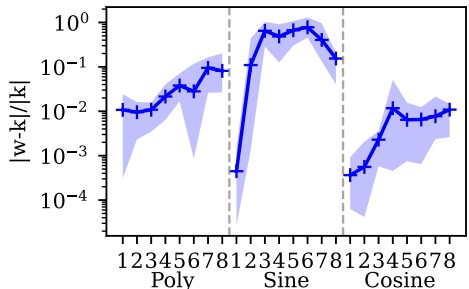

*Figure 3.* Learning a parameter using a PINN inverse problem (mean of five runs; shaded region is the min–max spread). Mirroring Theorem 3.1, the error grows with the polynomial degree $p$ of the training data, as the truncation error of the PDE loss is absorbed into the learned $w$.

The proof is in App. A.1, and empirically verified in Section 4.1. In other words, the error is irreducible, even with noiseless and infinite data regime, and is set entirely by the stencil order. Worse, enriching the data *hurts*: each polynomial degree past $q$ adds another $\mathcal{O}(\Delta x^m)$ term to the bias of $w$. This contradicts the classical ML intuition that richer training data should only help.

### 3.3. Training Dynamics for a Linear Model

Suppose now that we wish to learn a matrix $W$ which converges to the action of a solution operator $A$. To determine the expected $A$, the Green's function needs to be discretized relative to the discrete measurement space. The continuous forcing function can be constructed by $f(x) = \sum_i f_i \psi_i(x)$ where $\psi_i$ is a basis of the grid discretization. The discrete solution operator is thus,

$$u_i = \sum_j \left[ \int_0^1 G(x_i, s)\psi_j(s)\mathrm{d}s \right] f_j \qquad (9)$$

where the matrix in brackets defines $A$. The MSE loss over the dataset for this model is

$$\mathcal{L} = \frac{1}{N} \sum_{n=1}^{N} \left\| W f^{(n)} - u^{(n)} \right\|_2^2 \qquad (10)$$

More generally, suppose the data is sampled uniformly over subspaces of $\mathbb{R}^n$ such that $B$ is the matrix of rank $p + 1$ of coefficients satisfying $f(x_i) = f_i = Bc$. For example, a polynomial function space can use the Vandermonde matrix $B_{ij} = [1, x_i, x_i^2 ... x_i^p]$. By (9), there exists a matrix $A$ such that all training and test data satisfies $u = Af$. The following result shows that out of distribution generalizations *cannot* be expected:

**Theorem 3.2.** *Applying gradient descent on training dataset $\{u^{(n)}, f^{(n)}\}$ sampled from a function space $\mathcal{F}_{train}$, the model weights $W$ will converge to projection of the true operator $A$ onto the subspace of $\mathcal{F}_{train}$. Let the forcing functions be sampled by $f^{(n)} = Bc^{(n)}$ where each component of $c$ is sampled independently with $\mathbb{E}[c_i] = 0$. Optimizing the loss (10) using gradient descent with an initial condition $W^0$ will converge to*

$$W^* = AUU^T + W^0(I - UU^T). \qquad (11)$$

*where $U$ is the left orthogonal basis of $B$ of shape $N_{grid} \times (p + 1)$.*

The proof is in App. A.3, and empirically verified in Section 4.1. The learned matrix will converge to the projection of the true operator $A$ onto the subspace of data $F$, $AUU^T$, plus an initial error/noise that is orthogonal to the subspace of the data. This result is irrespective of the amount of data and the discretization, and as such, is actually a quite pessimistic result. In particular, $A$ is only learned iff the dimension of the subspace of $f$ is of equal rank with $A$. The finer the discretization, the greater the "quality" of the data needed (e.g., the dimension of the subspace).

# 4. Experiments

We examine eight different types of ML models in different settings: parameter-fitting inverse problems where the DE is known (finite difference, PINN), black-box models (linear, deep linear, multilayer perceptron), SciML black-box models (DeepONet, Neural Operator), and physics-informed learning models (Physics-Informed DeepONet).

A total of 25 different datasets are constructed using the function spaces described in Section 3.1: one piecewise linear dataset (FEM) and $p \in [1, 8]$ for polynomial, sine, and cosine spaces. For each choice of $(type, p, N_{\text{grid}})$, we generate 10,000 examples to represent the infinite data limit. Each dataset is normalized to have unit norm in $u$ to (1) minimize the effects of numerical ill-conditioning and round-off errors in the $p \geq 5$ datasets with extreme magnitudes, and (2) make the MSE comparable across datasets.

Our goal for the experiments is threefold: to show empirically that the theoretical a priori estimates from Section 3 hold (Sec. 4.1); to extend the results to architectures and problems where we do not have theory (Sec. 4.2); and to answer why Fig. 1 shows a weight matrix that looks nearly right yet inverts incorrectly (Sec. 4.3). The results are replicated in the presence of noise and on different linear PDEs. App. B details software, hardware, and hyperparameters, and App. D contains additional results on grid size variation.

## 4.1. Empirical Verification of Theoretical Predictions

The theoretical results of Section 3 can be used to compute predicted optimal parameters, allowing for a priori estimation of errors.

**Linear model.** Theorem 3.2 predicts that the converged weights are $\boldsymbol{W}^* = \boldsymbol{A}\boldsymbol{U}\boldsymbol{U}^T$ when $\boldsymbol{W}^0 = 0$, so the predicted MSE floor on the training distribution is the basis-projection residual $\|\boldsymbol{A} - \boldsymbol{A}\boldsymbol{U}\boldsymbol{U}^T\|_F^2$, computed from the SVD of the basis-evaluation matrix as described in App. A.4. The prediction matches observations nearly to machine precision for low $p$, and only diverges once $p$ is large enough that the basis matrix $\boldsymbol{B}$ becomes ill-conditioned (Fig. 2, left).

**Finite-difference model.** Theorem 3.1 gives a closed form for $w$ once the truncation coefficients of the stencil are fixed (App. A.2), producing the predicted curves overlaid on the measured errors in Fig. 2 (center, right). The prediction reproduces the empirical *slope* in $\Delta x$ and $p$ but overestimates the absolute magnitude: the bound in Theorem 3.1 treats the high-order truncation coefficients $\mu_m$ as worst-case constants, whereas the actual stencil error is typically smaller.

**Extension to PINNs.** The same mechanism appears when the explicit finite-difference stencil is replaced by a PINN with a PDE loss, implemented via the DeepXDE library (Lu et al., 2021b). While vanilla PINNs are used as numerical solvers, the PDE loss can be coupled with unknown parameters to solve an inverse problem:

$$\min_{\theta, w} \sum_{x \in grid} \left( w \frac{d^2}{dx^2} N(\theta, x_i) + f(x_i) \right)^2 + (N(\theta, x_i) - u_i)^2 \tag{12}$$

where $\theta$ and $w$ are learned simultaneously with $N(\theta, x_i) \approx u(x_i)$. Fig. 3 shows the parameter error $|w - k|/|k|$ grows with the polynomial degree of the training data, matching the finite-difference behavior by the same mechanism as Theorem 3.1.

## 4.2. OOD Dataset Generalization

Theorems 3.1 and 3.2 sharply characterize two specific parameter-learning settings. We now ask whether the same function-space dependence persists in operator-learning models for which we have no analogous theory.

We train models from scratch on datasets with $N_{\text{grid}} = 22$ using each of the 25 different datasets. After training, each model is evaluated on the 24 other datasets unseen during training. This results in a $25 \times 25$ grid of MSEs for each model, shown in Figs. 4–8 as heatmaps. The model with the best training MSE from 5 runs is chosen to plot the corresponding row in the heatmap. Error bars over multiple seeds are shown in App. C.

**Finite Difference Model** As predicted by Theorem 3.1, the test error observed in Fig. 4a increases both when testing on higher-order functions (moving right in the heatmap) and when training on higher-order functions (moving down in the heatmap). The FEM dataset error is orders of magnitude higher as it violates smoothness requirements for finite differences. See Fig. 13 in App. D for further analysis on $\Delta x$ terms.

**Linear Model** We turn to Theorem 3.2 to empirically verify its predictions of $\boldsymbol{W}$ in Fig. 4b. We initialize $W_0 = 0$ and do not use regularization. For each training run, the test error rises sharply when the test data are no longer in a subspace of the training data. There are three clear lower-triangular blocks in the heatmap representing the three families of nested subspaces, providing evidence for generalization across function subspaces. Increasing polynomial orders decrease the MSE on the cosine and sine datasets (forming pyramid structures in the top middle and right.) The FEM piecewise linear function space along the top row shows consistently low MSE in all models.

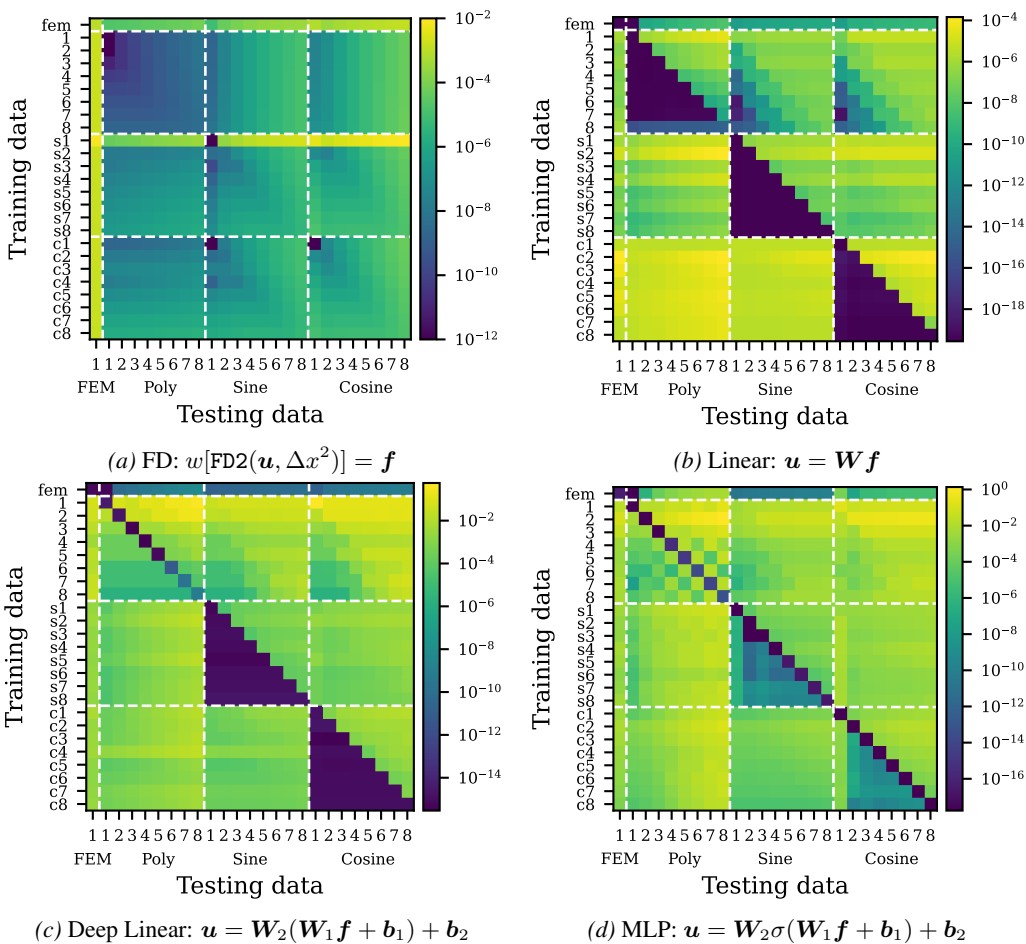

*(a)* FD: $w[\mathtt{FD2}(\boldsymbol{u}, \Delta x^2)] = \boldsymbol{f}$

*(b)* Linear: $\boldsymbol{u} = \boldsymbol{W}\boldsymbol{f}$

*(c)* Deep Linear: $\boldsymbol{u} = \boldsymbol{W}_2(\boldsymbol{W}_1\boldsymbol{f} + \boldsymbol{b}_1) + \boldsymbol{b}_2$

*(d)* MLP: $\boldsymbol{u} = \boldsymbol{W}_2\sigma(\boldsymbol{W}_1\boldsymbol{f} + \boldsymbol{b}_1) + \boldsymbol{b}_2$

*Figure 4.* Cross-evaluation error heatmaps for four model types. Rows: training set; columns: test set. Dashed lines separate function class families; cells left of each cell are subspaces. Color shows log-scale MSE (sharp transitions $\approx \times 10^{10}$). The blue lower triangular blocks in 4b and 4c indicate generalization with $\mathcal{L}(w, \mathcal{F}_{test}) \lesssim \mathcal{L}(w, \mathcal{F}_{train})$ when $\mathcal{F}_{\text{test}} \subseteq \mathcal{F}_{\text{train}}$.

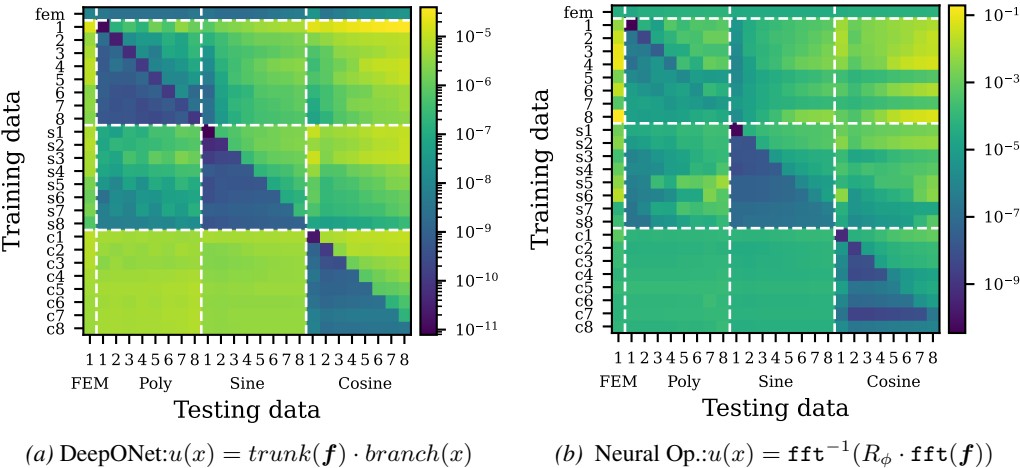

*(a)* DeepONet: $u(x) = trunk(\boldsymbol{f}) \cdot branch(x)$

*(b)* Neural Op.: $u(x) = \mathtt{fft}^{-1}(R_\phi \cdot \mathtt{fft}(\boldsymbol{f}))$

*Figure 5.* Generalization to out-of-distribution test datasets for the DeepONet and Fourier Neural Operator. The generalization patterns are similar to the Deep Linear models in Fig. 4.

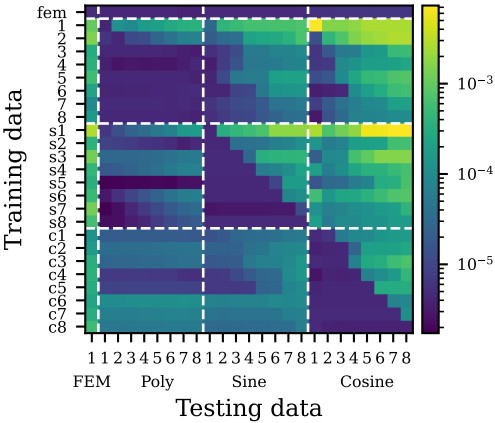

*Figure 6.* The generalization pattern for the Physics-Informed DeepONet is similar to the Deep ONet in Fig. 5, but with a higher baseline error on the training data.

**Deep Linear Models and Shallow Neural Networks** We train two basic nonlinear models: a deep linear model and a shallow neural network. As seen in Figs. 4c and 4d, neither model exhibits generalization behavior consistently. The MLP is strongly diagonal, showing that the model will over-fit to the training data, and not generalize in any case. The Deep Linear model shows mixed behavior: it exhibits sub-space generalization on the sine and cosine datasets but not on the polynomial datasets. As with the linear model, some cross-subspace generalization is observed when the models are trained with piecewise linear functions.

**DeepONet and Neural Operators** We investigate two models that are designed for learning PDEs: the DeepONet (Lu et al., 2021a) and the Fourier Neural Operator (Kovachki et al., 2023). The results are displayed in Fig. 5. Both models support flexible $x$-coordinate distributions; testing in our experiments is still performed on the regular grid for consistency. The DeepONet shows block lower-triangular structure similar to that of the linear model. The diagonals (evaluation on the training distribution) exhibit lower error values, showing a slight overfitting that does not strictly satisfy the subspace generalization. The dips in the trend are clear in Fig. 11e in App. C, where the model could satisfy $\mathcal{L}_{test} \lesssim \mathcal{L}_{train}$ tolerance depending on the application. The Neural Operator has a similar generalization pattern, but did not achieve a low training MSE on all function classes.

**Physics-Informed DeepONet** We implement a PI-DeepONet following (Wang et al., 2021) (Fig. 6), which

uses the loss:

$$
\min_{\theta} \sum_{f \in data} \sum_{x \in grid} \left( -k \frac{d^2}{dx^2} N(\theta, \mathbf{f}, x_i) - f(x_i) \right)^2
$$
$$
+ \sum_{x \in boundary} (\text{BC Loss}(N(\theta, \mathbf{f}, x_i)))^2 \quad (13)
$$

where $N(\cdot)$ is the factorized DeepONet that takes the observation point $x_i$ and the forcing function values $\mathbf{f}$ as inputs. Boundary conditions are enforced through an additional loss term. The baseline error is larger than the other models, achieving only $10^{-6}$ error. The subspace generalization pattern is still evident, although less pronounced due to the raised floor of the error, showing that physics informed losses do not mitigate the subspace limitation.

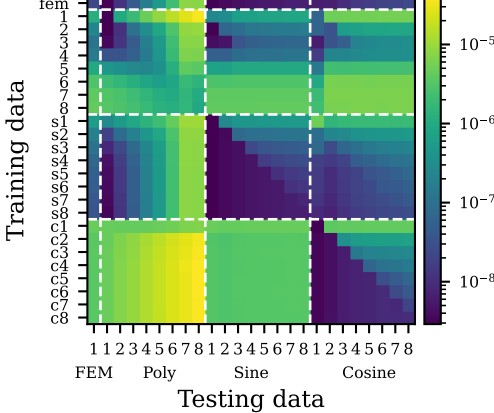

*Figure 7.* Poisson equation with training data corrupted by noise.

**Effect of Noise** We investigate the effect of measurement noise on the generalization boundaries by adding normally distributed noise to the 1D Poisson equation training data of $u$ and $f$. The results are displayed in Fig. 7. Rather than "smearing" the sharp generalization boundaries, noise raises the floor of achievable losses, achieving a lowest error of $\approx 10^{-9}$ compared to $10^{-20}$ error, and the generalization pattern is still present.

**2D Poisson and Biharmonic Equations** We repeat the same experiments for the 1D biharmonic and 2D Poisson equations using a linear model, shown in Fig. 8. The biharmonic equation, $-k \frac{d^4 u}{dx^4} = f$, displays the same block lower-triangular structure as the 1D case. Solutions for the biharmonic are constructed the same way as for the 1D Poisson equation. For the 2D Poisson equation, forcing functions are constructed using independent basis functions in $x$ and $y$, $f(x, y) = \sum_{i=1}^{p} c_i \sin(\pi i x) \sum_{j=1}^{q} d_j \sin(\pi j y)$. This exhibits subspace generalization independently in each spatial direction, producing the Sierpiński-triangle-esque pattern. This shows that the results hold generally for linear PDEs, as implied by Theorem 3.2.

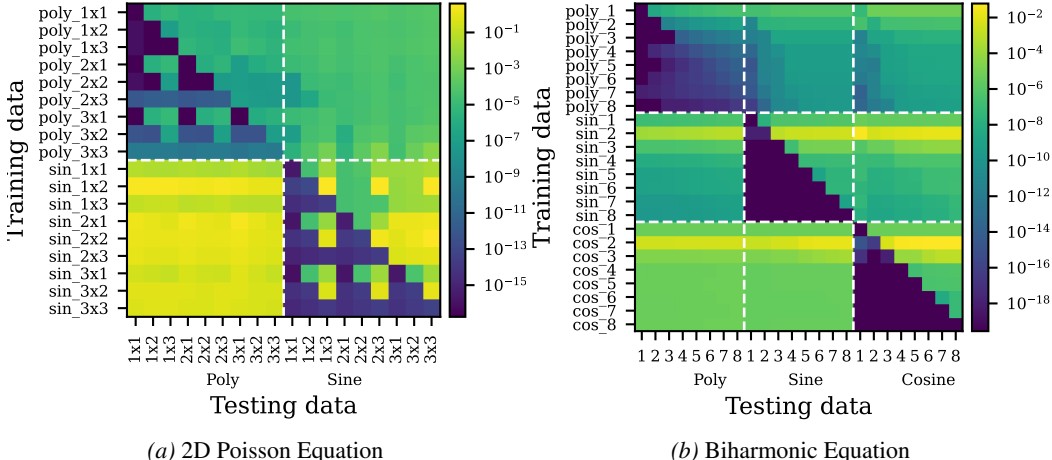

*(a)* 2D Poisson Equation           *(b)* Biharmonic Equation

*Figure 8.* Generalization to out-of-distribution test datasets constructed for the 2D Poisson and Biharmonic equations learned with a linear model.

### 4.3. Extracting Green's Functions From Black-Box Models

The Green's function is defined as the impulse response of the solution operator. Given a trained model, the action of the Green's function can be approximated by applying a test function $\hat{f}_k = \delta_{kj} = \boldsymbol{e}_j$ (the one-hot vector at index $j$) to extract the structure using the relation

$$\boldsymbol{A}_{ij} \leftrightarrow Model(\boldsymbol{f} = \boldsymbol{e}_j)_i. \tag{14}$$

For the linear model, this is the same as inspecting the weight matrix, $\boldsymbol{A}_{ij} \leftrightarrow \boldsymbol{W}_{ij}$. Fig. 9 elaborates on Fig. 1 to show the convergence of the weight matrix to the derived analytical solution for increasing modes (or complexity) of data. For lower orders $p$, the extracted functions are inscrutable, even though the model achieves low error on the training set. Training the linear model on the FEM dataset is sufficient to recover an accurate Green's operator $\boldsymbol{A}$. It is even possible to invert the weight matrix ($\boldsymbol{W}^{-1} = \hat{\boldsymbol{L}}$) to uncover a local stencil structure, which is evident in its tridiagonal appearance.

Fig. 10 shows that it is also possible to extract the Green's functions for deep models by evaluating the nonlinear models on one-hot inputs $\boldsymbol{f} = \mathbf{e}_j$. For $p = 3$, there is no apparent structure, even though the model achieves low MSE on the training set. The expected structure is extracted using the MLP and DeepONet when they are trained on FEM data. The structure is somewhat evident for the Fourier Neural Operator with the $cos[7]$ training run, but no others.

## 5. Conclusion

Learning spatially dependent problems is limited by the underlying function space of the training data and the observation discretization. We theoretically proved that this barrier exists on the simplest problem — the Poisson problem — on parameter fitting finite differences and linear models. We illustrated that carefully choosing the training data can mitigate this issue and that it is possible to generalize to other function spaces, which can inform data collection and cross-validation techniques. As another form of validation, black-box models can be directly interpreted as integrated Green's functions when they generalize effectively across many test domains. However, this is still pessimistic and presents a headwind for discovering unknown physics from real world data, given the inherent limitations in real world data collection. Confoundingly, different types of ML models can exhibit different and sometimes opposing generalization behaviors.

We demonstrated this behavior across a wide range of representative model types, from simple parameter fits to models designed for PDE learning. Physics-informed losses which incorporate prior knowledge (e.g., PINNs and Physics-Informed DeepONets) suffer the same subspace limitation as pure black-box models. Our analysis suggests that even the strongest physical priors may not mitigate the impact of the training data distribution.

While our theory required a priori knowledge of how the dataset was constructed, our results suggest that extracting generalized scientific knowledge may require new methods capable of compensating for these limitations by discovering the underlying subspace of the data. We propose this methodology to be an evaluation benchmark on future development of learning physics. As we have seen that even linear DEs can pose a challenge, we recommend that future proposals for DE learning techniques measure cross-set generalization.

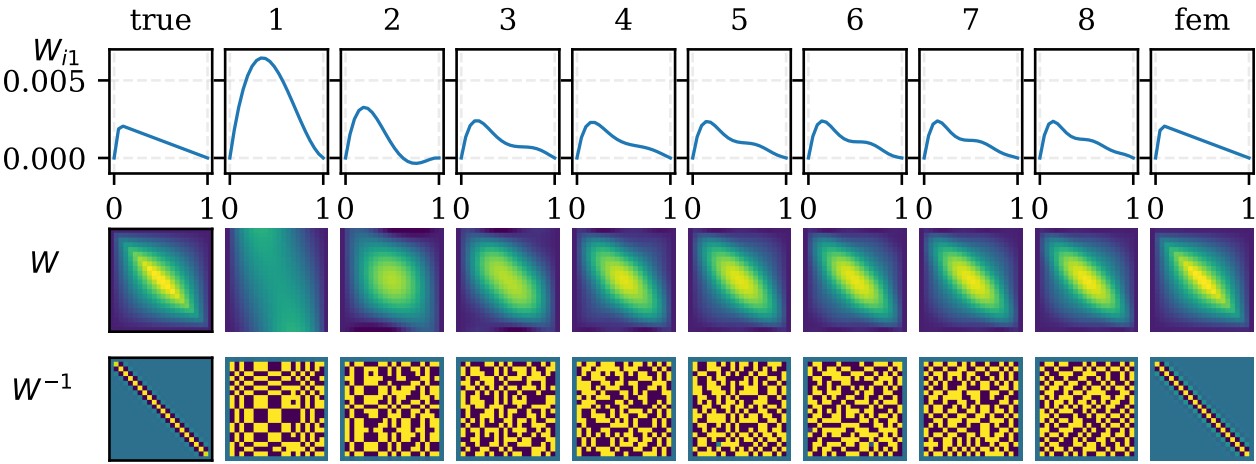

*Figure 9.* Visualization of the parameter matrix $W$ of the linear model as the dataset order increases. The top row displays one row $W_{i1}$ of the matrix, the row vector corresponding to the impulse at point $x = \Delta x$. It can be seen to converge to the shape of a Green's function. The middle row shows the entire matrix as a heatmap, and the bottom displays its inverse. The learned matrix converges incrementally towards the expected matrix (visualized from left to right) as terms are incrementally added to the training dataset. When the model is trained on the FEM dataset, it is possible to invert the learned $W$ and yield a banded structure of a local stencil. Parameters learned on all other dataset types cannot be inverted reliably.

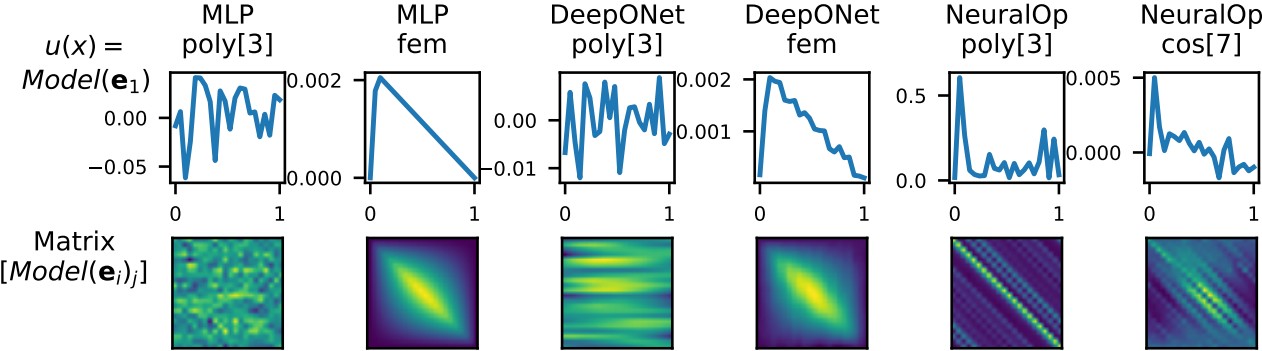

*Figure 10.* Visualization of test function evaluations, $\text{Model}(e_i)$, on black-box models can reveal Green's function structure (bottom row). The top row shows one test function evaluation at the node $x = \Delta x$, and the bottom row shows a matrix constructed from model evaluations for all test functions. The colormap scale matches the y-scale of the corresponding plot above them.

## 6. Limitations

Our theoretical analysis is restricted to linear PDEs; extending the convergence results to nonlinear settings requires additional machinery and is left for future work. The cross-validation methodology is directly applicable beyond the linear setting, but the experimental validation here covers only the 1D Poisson, biharmonic, and 2D Poisson equations. We do not propose a remedy for the identified generalization failure — the goal of this work is to characterize and measure the problem, not to solve it.

## Impact Statement

This paper analyzes limitations in discovering underlying physics from data across an entire spectrum of modeling approaches, with implications for evaluation and validation in scientific ML. These findings are relevant for applications in physics and engineering, where understanding out-of-distribution behavior is important for safe real-world deployment. By providing rigorous criteria for when a model has genuinely learned the underlying physics versus merely interpolated within its training subspace, this work could help practitioners avoid costly retraining cycles and unnecessary large-scale compute expenditure. The cross-validation methodology introduced here is lightweight to apply and could serve as an inexpensive early diagnostic before committing to expensive training runs or deployment. The work does not introduce new capabilities with direct societal risk.

## Acknowledgments

This paper describes objective technical results and analysis. Any subjective views or opinions that might be expressed in the paper do not necessarily represent the views of the U.S. Department of Energy or the United States Government.

This article has been authored by an employee of National Technology & Engineering Solutions of Sandia, LLC under Contract No. DE-NA0003525 with the U.S. Department of Energy (DOE). The employee owns all right, title and interest in and to the article and is solely responsible for its contents. The United States Government retains and the publisher, by accepting the article for publication, acknowledges that the United States Government retains a non-exclusive, paid-up, irrevocable, world-wide license to publish or reproduce the published form of this article or allow others to do so, for United States Government purposes. The DOE will provide public access to these results of federally sponsored research in accordance with the DOE Public Access Plan https://www.energy.gov/downloads/doe-public-access-plan. The work performed at Sandia National Laboratories was supported by the U.S. Department of Energy, Office of Science, Office of Advanced Scientific Computing Research, DyGenAI project, and the SEA-CROGS project in the MMICCs program. Additional support was received from Interlab Laboratory Directed Research and Development program at Sandia.

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

## A. Proofs

### A.1. Proof of Theorem 3.1

*Proof.* The training forcing function and corresponding analytical solution is given by

$$f^{(n)}(x) = -k \sum_{m=0}^{p} c_m x^m \tag{15}$$

$$u^{(n)}(x) = \sum_{m=-2}^{p} \frac{c_m x^{m+2}}{(m+2)(m+1)} \tag{16}$$

for coefficients $c_m$. Recall that a finite difference stencil of accuracy order $q$ will satisfy

$$FD_q(u, \Delta x) = u'' + \mathcal{O}(\Delta x^q). \tag{17}$$

Thus for a polynomial whose true derivative is $\sum c_m x^m$, the finite difference stencil can be written as a polynomial whose coefficients change starting at its accuracy order $q$,

$$FD_q(u, \Delta x) = \sum_{m=0}^{q-1} c_m x^m + \sum_{m=q}^{p} c_m (1 + \zeta_m \Delta x^m) x^m \tag{18}$$

$$:= \sum_m b_m x^m \tag{19}$$

where $\zeta_m$ is a bound of the coefficient of error introduced for each term $m$ starting at $q$ by Taylor's theorem. These error terms are dependent upon the finite difference stencil; the following section illustrates two examples. This proof is for the general case. For the usual 3-point stencil, $q = 2$. Simplifying the loss

$$\mathcal{L} = \mathbb{E}_c \left[ \int_0^1 \left( w \sum_{m=0}^{p} b_m x^m - k \sum_{m=0}^{p} c_m x^m \right)^2 dx \right] \tag{20}$$

$$= \mathbb{E}_c \left[ \int_0^1 \left( \sum_{m=0}^{p} (w b_m - k c_m) x^m \right)^2 dx \right] \tag{21}$$

where $b_m = c_m$ for $m < q$ and $b_m = c_m(1 + \zeta_m \Delta x^m)$ for $m \geq q$. Expanding the square, applying the integral and simplifying using the definition of $b_m$

$$\mathcal{L} = \mathbb{E}_c \left[ \int_0^1 \left( \sum_{j=0}^{p} \sum_{m=0}^{p} (w b_m - k c_m)(w b_j - k c_j) x^{m+j} \right) dx \right] \tag{22}$$

$$= \mathbb{E}_c \left[ \sum_{j=0}^{p} \sum_{m=0}^{p} \frac{(w b_m - k c_m)(w b_j - k c_j)}{m + j + 1} \right] \tag{23}$$

$$= \mathbb{E}_c \left[ \sum_{j=0}^{p} \sum_{m=0}^{p} \frac{(w(1 + \xi_m) - k)(w(1 + \xi_j) - k) c_m c_j}{m + j + 1} \right] \tag{24}$$

$$= \sum_{j=0}^{p} \sum_{m=0}^{p} \frac{(w(1 + \xi_m) - k)(w(1 + \xi_j) - k) \mathbb{E}_c [c_m c_j]}{m + j + 1} \tag{25}$$

where we let $\xi_m := \zeta_m \Delta x^m$. Since $c_m$ are iid, then $\mathbb{E}[c_m c_j] = \mathbb{E}[c_m^2]\delta_{mj}$. This lets us only consider the diagonal terms of the sum,

$$\mathcal{L} = \sum_{m=0}^{p} (w(1 + \xi_m) - k)^2 \frac{\mathbb{E}_c [c_m^2]}{2m + 1}. \tag{26}$$

Taking the derivative with respect to $w$

$$\frac{d\mathcal{L}}{dw} = \sum_{m=0}^{p} 2(1 + \xi_m)(w(1 + \xi_m) - k)\frac{\mathbb{E}_c\left[c_m^2\right]}{2m + 1}. \tag{27}$$

Setting $d\mathcal{L}/dw = 0$, the minima is at

$$w = \frac{\sum_{m=0}^{p}(1 + \xi_m)\frac{\mathbb{E}_c[c_m^2]}{2m+1}}{\sum_{m=0}^{p}(1 + \xi_m)(1 + \xi_m)\frac{\mathbb{E}_c[c_m^2]}{2m+1}}k. \tag{28}$$

By assumption that the expectations on the second moment are identical for each coefficient $c_m$, we can simplify and then break up the sum,

$$w = \frac{\sum_{m=0}^{p}(1 + \xi_m)\frac{1}{2m+1}}{\sum_{m=0}^{p}(1 + \xi_m)(1 + \xi_m)\frac{1}{2m+1}}k \tag{29}$$

$$= \frac{\sum_{m=0}^{q-1}\frac{1}{2m+1} + \sum_{m=q}^{p}(1 + \zeta_m \Delta x^m)\frac{1}{2m+1}}{\sum_{m=0}^{q-1}\frac{1}{2m+1} + \sum_{m=q}^{p}(1 + 2\zeta_m \Delta x^m + \zeta_m^2 \Delta x^{2m})\frac{1}{2m+1}}k \tag{30}$$

Now by algebraically using this to compute the error term, we have

$$\frac{w - k}{k} = \frac{-\sum_{m=q}^{p}(\zeta_m \Delta x^m + \zeta_m^2 \Delta x^{2m})\frac{1}{2m+1}}{\sum_{m=0}^{q-1}\frac{1}{2m+1} + \sum_{m=q}^{p}(1 + 2\zeta_m \Delta x^m + \zeta_m^2 \Delta x^{2m})\frac{1}{2m+1}}. \tag{31}$$

Note that $w = k$ if $p < q$ as the numerator vanishes. Conversely, when $p > q$, $w \to k$ as $\Delta x \to 0$ at a rate of $\Delta x^q$ by polynomial division. Furthermore, for a fixed $\Delta x$, increasing $p$ increases the error. Each additional term adding $p$ adds additional terms: at the very least, the term $\xi_m \Delta x^{2m}$ is always positive, so the magnitude of that term will always increase, even if $\zeta_m \Delta x^m$ has favorable canceling out.

Thus, the dominant terms are in the numerator,

$$\frac{w - k}{k} = \sum_{m=q}^{p}(\mu_m \Delta x^m) \tag{32}$$

where the particular coefficients $\mu_m$ are finite constants that can be derived for particular finite difference rules. $\qquad\square$

### A.2. Concrete Example of Error for 3-Point Stencil

For a given stencil, the above procedure can be executed algebraically to produce the graph in Fig. 2 (center, right). Plugging in the finite difference stencil into Equation (7), the inner value of the expression is

$$\frac{w}{\Delta x^2}\left(\left(\sum_{n=0}^{p+2} c_n(x - \Delta x)^n\right) - 2\left(\sum_{n=0}^{p+2} c_n x^n\right) + \left(\sum_{n=0}^{p+2} c_n(x + \Delta x)^n\right)\right) - k\sum_{n=2}^{p+2} n(n-1)c_n x^{n-2} \tag{33}$$

For $p = 0$ and $p = 1$, the equation collapses to $w = k$. But when we consider $p = 2$ with quadratic $f$ and quartic $u$, there is not full cancellation:

$$\mathcal{L} = \mathbb{E}_c\left[\int_0^1\left(2w\left(c_0 + 3c_1 x + 6c_2 x^2 + c_4 \Delta x^2\right) - 2k\left(c_0 + 3c_1 x + 6c_2 x^2\right)\right)^2 dx\right] \tag{34}$$

Following the procedure in the proof algebraically, we have

$$w = \frac{3k\left(105\Delta^2 + 223\right)}{245\Delta^4 + 630\Delta^2 + 669} \tag{35}$$

which leads to an error term

$$\frac{w - k}{k} = \frac{\Delta^2\left(-245\Delta^2 - 315\right)}{245\Delta^4 + 630\Delta^2 + 669} \tag{36}$$

which satisfies the general result and was used to plot the graph in Fig. 2 (center). A similar algebraic equation can be derived for the five point stencil to yield the graph in Fig. 2 (right).

### A.3. Proof of Theorem 3.2

*Proof.* By construction of our dataset and ansatz on $A$, the $n$th sample is $f^{(n)} = Bc^{(n)}$ and $u^{(n)} = ABc^{(n)}$, thus the loss is

$$\mathcal{L} = \frac{1}{2N} \sum_{n=1}^{N} ||ABc^{(n)} - WBc^{(n)}||^2 \tag{37}$$

where $A, W$ are of size $N_{\text{grid}} \times N_{\text{grid}}$, $c$ is of size $p+1$ and $B$ is size $N_{\text{grid}} \times (p+1)$.

Clearly $W = A$ is a solution, but if $B$ is lower rank than $A$ and $W$, there is a nullspace in the least squares equation. Gradient descent moves in the direction of the gradient of the loss

$$\frac{\partial \mathcal{L}}{\partial W} = -\frac{1}{N} \sum_{n=1}^{N} (A - W)(Bc^{(n)}(c^{(n)})^T B^T). \tag{38}$$

With the assumption of iid $c$, then $\mathbb{E}[c_m c_k] = \mathbb{E}[c_m^2]\delta_{mk}$. For any sampling and basis set, we can rescale $B_{:,m}$ such that $\mathbb{E}[c_m^2] = 1$ to simplify the following. By linearity of expectation, we have that the expected gradient is simply

$$\frac{\partial \mathcal{L}}{\partial W} = -(A - W)BB^T. \tag{39}$$

The minima will satisfy

$$0 = (A - W)BB^T \tag{40}$$

via the dynamics

$$W_{t+1} = W_t + \eta(A - W_t)BB^T. \tag{41}$$

With initial condition $W^0$, we will show that the sequence converges to

$$W^* = AUU^T + W^0(I - UU^T) \tag{42}$$

where $U$ is the left $N_{\text{grid}} \times (p+1)$ eigenvector matrix of the SVD decomposition $B = UDV^T$. Define the error at each step $t$ as

$$E_t = W_t - W^* \tag{43}$$

The error can be decomposed into two components. $E^{||}$ which is in the row space of $BB^T$ and $E^{\perp}$ which is in the null space $null(BB^T) = I - UU^T$. The matrix $BB^T = UDV^TVD^TU^T = UD^2U^T$. Since the gradient update is always in the row space of $BB^T$ this implies that $E^{\perp}$ is constant 0 as the following calculation shows

$$\begin{aligned}
E_0^{\perp} &= W_0^{\perp} - W^{*\perp} \\
&= W_0(I - UU^T) - (AUU^T + W_0(I - UU^T))(I - UU^T) \\
&= W_0(I - UU^T) - W_0(I - UU^T) = 0
\end{aligned}$$

where we used $UU^T(I - UU^T) = 0$. Thus, only considering the component parallel to $UU^T$, we perform the algebra on the updates to the error,

$$\begin{aligned}
E_{t+1}^{||} &= W_{t+1}^{||} - W^{*||} = (W_{t+1} - W^*)UU^T \\
&= (W_{t+1} - AUU^T - W_0(I - UU^T))UU^T \\
&= W_{t+1}UU^T - AUU^T \\
&= (W_t + \eta(A - W_t)BB^T)UU^T - AUU^T \\
&= W_t U(I - \eta D^2)U^T - AU(I - \eta D^2)U^T
\end{aligned}$$

Assuming the singular values of $B$ are identical,

$$\begin{aligned}
&= (1 - \eta D^2)(W_t^{||} - W^{*||}) \\
&= (1 - \eta D^2)E_t^{||}.
\end{aligned}$$

Thus, for any $\eta < 1/\max(D^2)$, the components parallel to $UU^T$ go to zero $E_t^{||} = (1 - \eta D^2)^t E_0^{||} \to 0$. As both error terms converge to 0, we have shown $W_t \to W^*$ as desired. $\qquad\square$

Corollary to Theorem 2: A Loose Error Bound A loose error bound naturally follows: Assuming that the basis set has $p + 1$ entries, the error $\|\boldsymbol{W}^* - \boldsymbol{A}\|_F / \|\boldsymbol{A}\|_F$ is dependent on the size of the basis set,

$$\frac{\|\boldsymbol{W}^* - \boldsymbol{A}\|_F}{\|\boldsymbol{A}\|_F} \leq \sqrt{N_{\text{grid}} - (p+1)} \left( \frac{\|\boldsymbol{W}^0\|_F}{\|\boldsymbol{A}\|_F} + 1 \right) \tag{44}$$

*Proof.* Note that, $\boldsymbol{W}^* - \boldsymbol{A} = \boldsymbol{W}^0(\boldsymbol{I} - \boldsymbol{U}\boldsymbol{U}^T) - \boldsymbol{A}(\boldsymbol{I} - \boldsymbol{U}\boldsymbol{U}^T) = (\boldsymbol{W}^0 - \boldsymbol{A})(\boldsymbol{I} - \boldsymbol{U}\boldsymbol{U}^T)$. The Frobenius norm satisfies the sub-multiplicative property, meaning $\|(\boldsymbol{W}^0 - \boldsymbol{A})(\boldsymbol{I} - \boldsymbol{U}\boldsymbol{U}^T)\|_F \leq \|\boldsymbol{W}^0 - \boldsymbol{A}\|_F \|\boldsymbol{I} - \boldsymbol{U}\boldsymbol{U}^T\|_F$. The matrix $\boldsymbol{U}\boldsymbol{U}^T$ is rank $p + 1$. The complementary matrix has rank $N_{\text{grid}} - (p + 1)$. The F-norm for a rank $r$ orthogonal projector is $\sqrt{r}$, since $\|\boldsymbol{P}\|_F = \sqrt{\text{tr}(\boldsymbol{P}^T\boldsymbol{P})} = \sqrt{\text{tr}(\boldsymbol{P})} = \sqrt{r}$. By the triangle inequality, $\|\boldsymbol{W}^0 - \boldsymbol{A}\|_F \leq \|\boldsymbol{W}^0\|_F + \|\boldsymbol{A}\|_F$. Assembling these gives the stated result. $\square$

### A.4. Corollary to Theorem 3.2: Error Estimate

If the Green's function and data basis is known, the analytical solution $\boldsymbol{A}$ can be computed to give estimate of error, shown in Fig.2. The procedure is as follows: Compute a matrix of basis functions evaluated at each point in the domain,

$$\boldsymbol{B}_{ij} := [\phi_0(x_i), \phi_1(x_i), \ldots, \phi_p(x_i)] \quad x_i = i\Delta x \tag{45}$$

and compute the SVD of the basis matrix:

$$\boldsymbol{U}, \boldsymbol{D}, \boldsymbol{V} := \text{SVD}(\boldsymbol{B}) \tag{46}$$

Generate the Green's function matrix by symbolically integrating the Green's function using a compact basis:

$$\boldsymbol{A}_{ij} = \int_\Omega \psi_i(s) G(s, x_j) \mathrm{d}x \tag{47}$$

where $\psi_i$ is a piecewise linear or piecewise constant basis function centered on $x_i$. Then, the error starting at any initial matrix $\boldsymbol{W}^0$ can be directly computed as

$$\|\boldsymbol{W}^* - \boldsymbol{A}\|_F \approx \|\boldsymbol{W}^0 - \boldsymbol{A}\boldsymbol{U}\boldsymbol{U}^T\|_F \tag{48}$$

where the computation for $\boldsymbol{A}$ is imperfect from the arbitrary choice of discretization basis $\psi(x)$.

## B. Procedure Details

The ultimate experiments ran in float32 on a MacBook Pro M3 Max GPU using PyTorch. To initially rule out errors from accelerator hardware, drivers, or numerical precision, we replicated the linear models and deep models in many implementations: using Tinygrad, JAX, and PyTorch; on CPU, NVIDIA 4070 GPU, and Apple Silicon GPU hardware; and varied precision in float32 and float64. The behavior was consistent across all hardware and software variations.

Finite difference experiments were performed using pysindy (Kaptanoglu et al., 2022); the non-temporal spatial problem was implemented by setting x_dot[i] = $\boldsymbol{f}_i$. The PINN, DeepONet, and Physics-Informed DeepONet were trained using the DeepXDE library (Lu et al., 2021b). The Neural Operator is trained using the neuralop library (Kossaifi et al., 2024). The piecewise linear datasets were generated using the finite element method using FEniCS (Baratta et al., 2023).

All models were trained using AdamW with weight decay $\lambda = 0.01$. The Linear and Deep Linear models used full-batch training on 10k examples for 2,000 epochs with a linear learning rate schedule from 1e-1 to 1e-6; the Deep Linear has one hidden layer of dimension 100. The MLP (1 layer, hidden dimension 1024, Leaky ReLU) and Fourier Neural Operator (64 modes, hidden dimension 32, GeLU) used batch size 256 for 5,000 epochs with a linear schedule from 1e-2 to 1e-6. The DeepONet used branch tower dimensions [1, 256, 256], trunk tower dimensions [$N_{\text{grid}}$, 256, 256], ReLU activation, batch size 256, and a step schedule from 1e-3 decaying to 1e-6 every 5,000 steps over 20,000 total steps.

## C. Repeatability of Test Set Generalization

Repeated experiments with different seeds are shown in Figures 11a, 11b, 11c, 11d, 11e, and 11f. Each line on this plot corresponds to training on one function class identified in the legend, which corresponds to a row in the heat map from Figures 4 and 5. The error bars represent min/max over 5 runs, and the line is the average of the five. The repeated runs all show the same qualitative trends of generalization and overfitting in the same regimes for each of the six types of models.

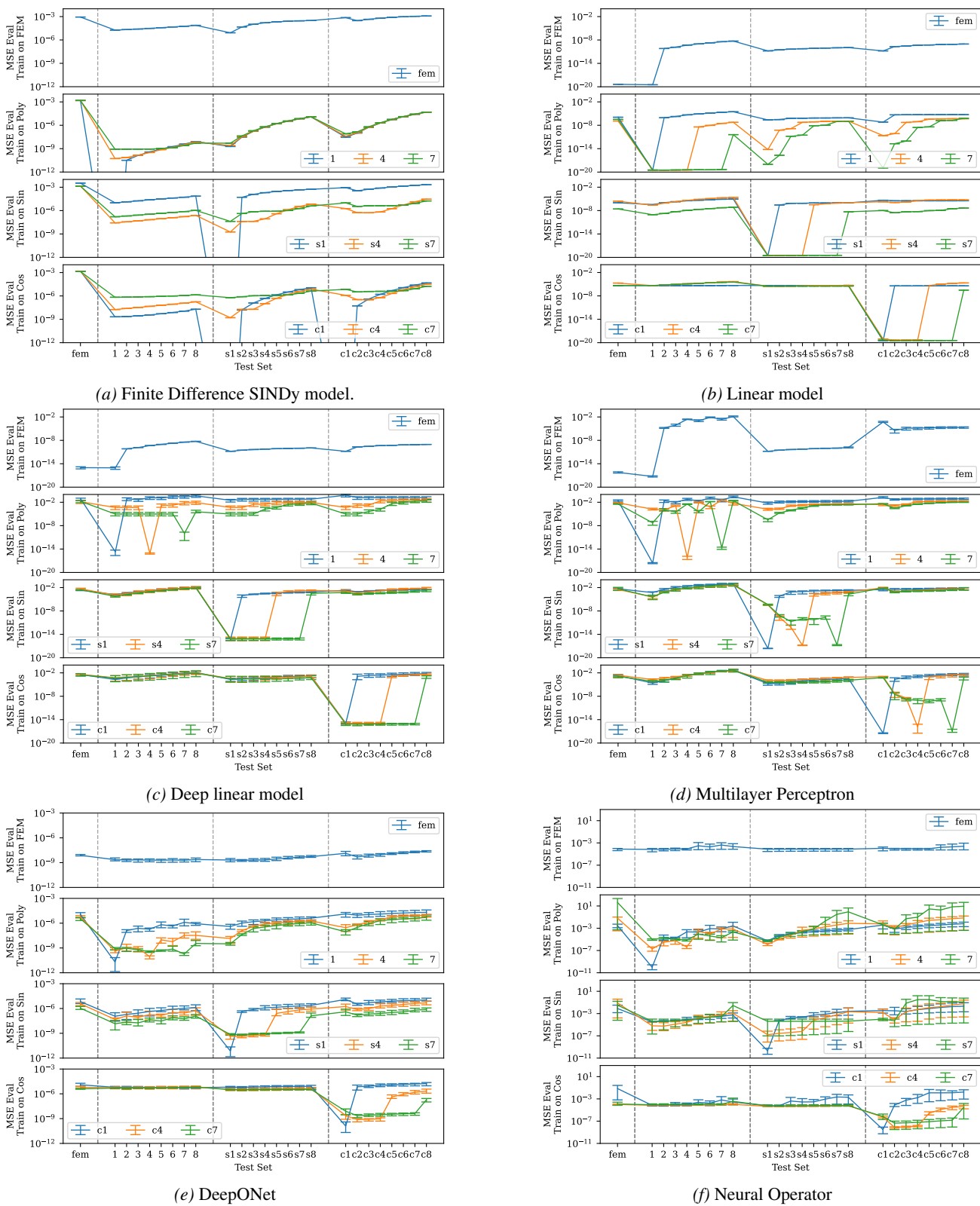

*Figure 11.* Generalization errors for all models with five different seeds for each training run. For the Finite Difference SINDy model, the lower y limit is truncated to $10^{-12}$; training on Poly 1, Sine 1 and Cosine 1 achieves error $\approx 10^{-38}$.

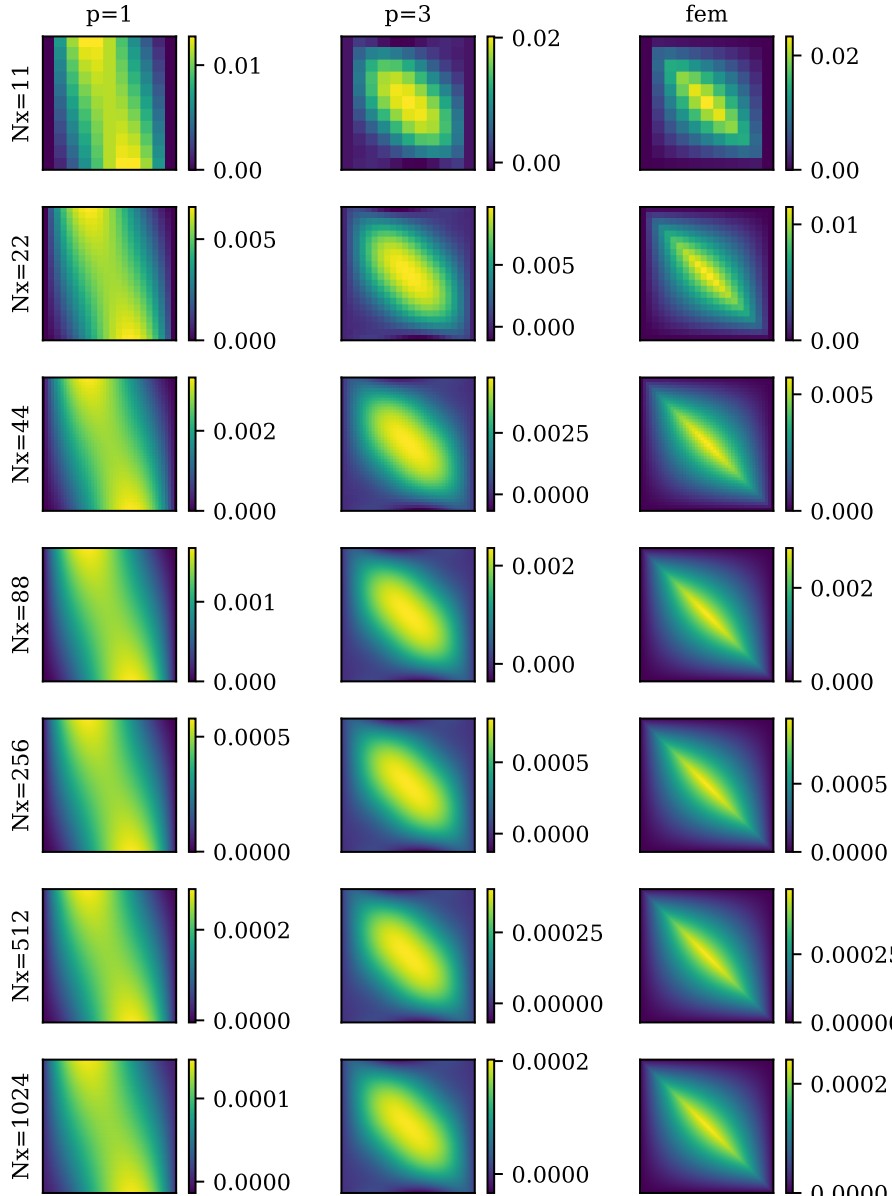

*Figure 12.* Convergence of the weight matrix $W$ as the grid size increases. The matrix structure is invariant to the discretization, however, each parameter scales proportional to $\Delta x$ (Eq. 9).

## D. Grid Size Generalization

We consider the situation where we train and evaluate models on datasets produced at different grid spacings. Note that $\Delta x$ is a property of the training data: decreasing $\Delta x$ in practice would require finer measurements. In Fig. 12, we observe that the shape of the matrix does not change as the grid size increases. The polynomial datasets have the same number of coefficients for each grid size. However, the rank of "fem" dataset does increase as the piecewise linear functions are sampled on the new grid. For the finite difference method in Fig. 13, we do observe a difference in the error. As predicted by Theorem 2, the error on the learned parameter is proportional to both the grid size and polynomial order of the dataset. The order of the stencil does change the rate of convergence towards the true parameter.

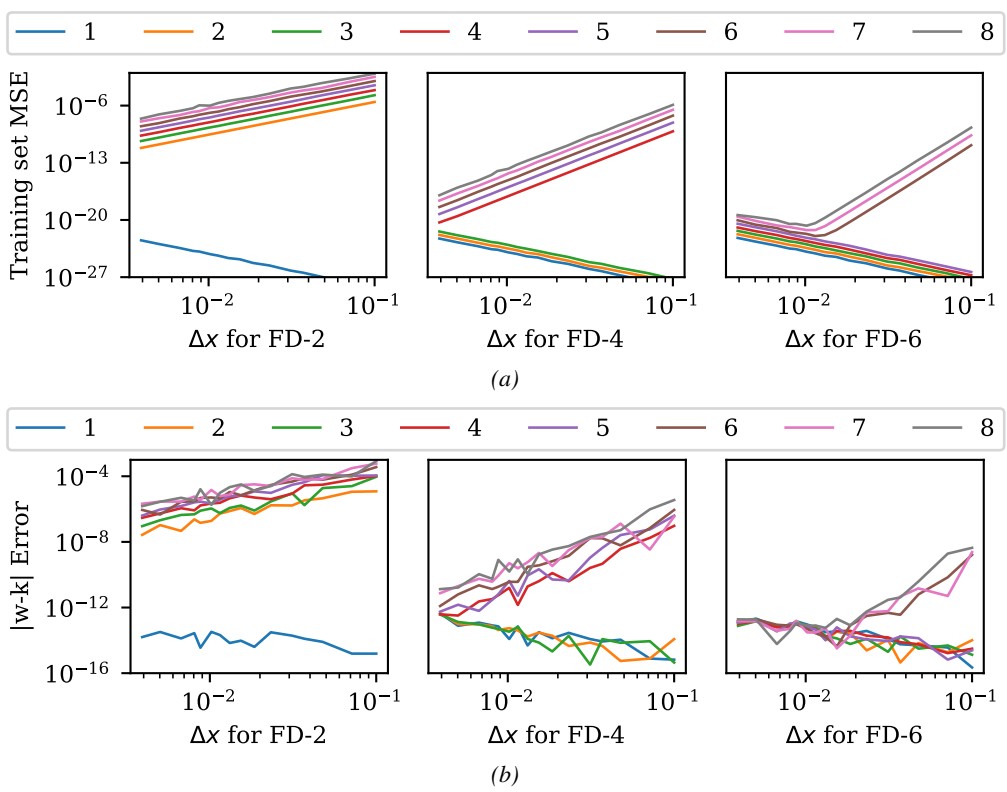

*Figure 13.* Experiments on SINDy equation discovery as the grid size (x-axis) and training data polynomial function order (line color in legend) is changed. 13a: training data MSE (observable by the scientist). 13b : difference from true parameter (not observable). After convergence, the error tends to increase; we believe this is an artifact of accumulation of floating point error.

