# OpenReview forum: "Interpretability and Generalization Bounds for Learning Spatial Physics"
_ICML.cc/2026/Conference — ICML 2026 regular_

### Official Review · Reviewer_ckh4 · 2026-03-02

**Soundness:** 3
**Presentation:** 3
**Significance:** 2
**Originality:** 2
**Overall Recommendation:** 4
**Confidence:** 2

**Summary:**

The paper points out that current machine learning methods for differential equations lack interpretability and generalization. Based on a simple Poisson equation, the authors show potential failure modes and possible fixes -- e.g., careful dataset construction.

**Compliance With Llm Reviewing Policy:**

Affirmed.

**Final Justification:**

The author's rebuttal addressed my concern regarding the applicability to nonlinear equations, hence I lean towards acceptance. The authors should still discuss clearly the applicability of their method, when it is advantageous to other methods and when it is not.

**Key Questions For Authors:**

* Can you discuss how your work is differentiated from neural networks that explicitly learn Green's function, e.g., "Deep Generalized Green’s Functions", "Multiscale Neural Networks for Approximating Green's Functions"?
* Can you explain how your conclusions could generalize beyond nonlinear equations?

**Limitations:**

Yes

**Strengths And Weaknesses:**

* The paper is not very sound. The writing seems to imply that the authors are ignorant of the literature. For example, I find the sentence in the abstract, "While there are many applications of machine learning (ML) to scientific problems that look promising, the eye test can be misleading compared to quantitative measures," quite confusing, since it is almost a trivial statement. For example, what is the "eye testing" that the authors intend to compare against?
* Presentation is overall good.
* Significance. This paper only demonstrates the Poisson equation. Since it's a linear equation, Green functions work, but I'm afraid the conclusions cannot generalize beyond linearity.
* Originality: The conclusions in this paper are not very novel. The setup is too simple.

---

> ### Author Rebuttal · Authors · 2026-03-31
>
> We thank the reviewer for their constructive feedback. We have addressed the main points as follows:
>
>
> **On the "eye test" phrasing.**
>
> We apologize for any confusion caused by our use of a colloquialism that is not universal.
> In the field of numerical simulation, the phrase "eye test" is used to describe comparing a plotted prediction against a reference solution by visual inspection as a means of verification (which is understood to be an insufficient verification).
> We invoked the "eye test" colloquialism to critique the tendency in the ML for physics literature to over-index on visual results and not meet the standards of numerical analysis.
> ML predictions on physical systems can appear qualitatively accurate (passing the proverbial eye test) while quantitative error analysis reveals that the results are insufficient at scientific and engineering precision.
> This is demonstrated concretely in Fig. 1: the learned matrix $W_{p=3}$ visually resembles the true Green's function operator $A$, but the inverse of this matrix reveals that the model has not learned the correct physics, and OOD errors can exceed in-distribution errors by factors of $10^8$ or more (Section 1, Eq. 5). We will reword this in the revision for precision while avoiding colloquialisms.
>
> **Differentiation from neural networks that explicitly learn Green's functions.**
>
> We appreciate this question, as it highlights an important distinction. We did cite works that learn Green's functions directly (Gin et al., 2021; Boullé et al., 2022; 2023), but omitted them in our analysis for brevity. Those papers use *architectures designed to learn Green's functions* while we strictly *extract* Green's function representations from the weights of *any black-box models* (MLP, DeepONet, FNO).
> We showed that this extraction succeeds only when the training data spans a sufficient subspace of the operator: we connect the interpretability with a new type of verification method.
> To our knowledge, no prior work has used probing to extract Green's function representations from models that were not architecturally designed to represent them, or connected the quality of this extraction to the coverage of training data.
> The reviewer does bring up an insightful connection to the learned Green's function methods, and we will note the connection in the text.
>
> **How conclusions generalize beyond linear equations.**
>
> Our empirical results on deep nonlinear architectures (MLPs, DeepONets, FNOs, PINNs) demonstrate that the subspace-dependent failure mode is a property of the learning procedure, not the equation.
> Moreover, the cross-validation methodology we propose (train on one function class, evaluate on others) is directly applicable to any PDE, linear or nonlinear, and any type of ML model.
> In any context, we pose that it can reveal whether a model has learned generalizable physics or merely interpolated within a training subspace.
> Our theoretical analysis also backs the new view that the training / validation splits need be constructed along subspaces or distinct regimes of the datasets to measure generalization, as opposed to purely randomized splits.
> We believe that the (train,eval) cross validation grid is an intuitive and easy to replicate validation technique that will be adopted and inform future development of SciML methods that can address the generalization failures.
>
> **On soundness.**
>
> We respectfully note that the review rates soundness as "poor" but does not identify a specific technical error in the theorems, proofs, or experimental methodology. Additionally, no other review identifies a concrete mathematical or experimental flaw. We believe the paper's claims are supported by the analysis and empirical evidence as written, and we would be glad to address any specific technical concern the reviewer may have.

---

> > ### Author Rebuttal · Reviewer_ckh4 · 2026-04-02
> >
> > Thanks for the clarification! I've raised my score to 4.

---

> > > ### Author Response · Authors · 2026-04-07
> > >
> > > We thank the reviewer for the engagement and for confirming that all concerns have been addressed.

---

### Official Review · Reviewer_cBuA · 2026-03-11

**Soundness:** 2
**Presentation:** 1
**Significance:** 2
**Originality:** 2
**Overall Recommendation:** 2
**Confidence:** 3

**Summary:**

The paper studies machine learning models applied to a simple linear ODE, the 1D Poisson equation $-k u'' = f$. The dataset consists of $(f, u)$ pairs and varies along two axes: (1) the grid spacing used for discretization, and (2) the function class from which $f$ is sampled. The paper first considers learning the constant $k$ and shows that the convergence rate of the learned parameter to the true $k$ depends on the grid spacing and the polynomial degree of the function class. It then analyzes learning the operator mapping $u \mapsto f$ as a linear regression problem, demonstrating that the model only captures the subspace spanned by the training basis and therefore fails to generalize OOD. Finally, it presents many empirical results with more complex neural network models, investigating the OOD generalization in those cases.

**Compliance With Llm Reviewing Policy:**

Affirmed.

**Final Justification:**

The authors' rebuttal successfully clarified their specific contributions, and I have increased the Originality score accordingly. However, I maintain my recommendation to Reject. While the observations regarding out-of-distribution (OOD) failure may be useful to practitioners, the theoretical results themselves appear standard from a theorist's perspective. More critically, the paper lacks a coherent narrative combining both theoretical and empirical perspective, and hence requires a substantial restructuring to deliver its message clearly. In its current state, the manuscript is not yet ready for publication as the lack of a focused narrative significantly hinders readability.

**Key Questions For Authors:**

See weakness.

**Limitations:**

Yes

**Strengths And Weaknesses:**

Strengths:
- The paper focuses on a simple, well-scoped setting that supports both theoretical and empirical analysis. I like the concrete OOD generalization questions in this case (i.e., shifting $f$ across function classes), leading to specific, testable claims.
- The overall claim is convincing and clear: ML models cannot learn the true physics in the simplest settings.

Weaknesses:
1. The writing does not present a clear, cohesive narrative. It makes several theoretical and empirical observations, but they all appear disjointed and do not coalesce into a central message that offers new insight. The key point, such as the lack of generalization across different function classes and the failure to recover underlying physics, are unsurprising for theoretist, and the paper does not provide novel perspectives on these issues.
2. While there is some theoretical analysis, the results largely follow from standard tools in numerical analysis and basic ML theory. Despite claims to the contrary, they do not seem particularly surprising or insightful, and the contribution to the literature appears limited. For such clear and simple set up (which actually gives linear regression), I believe much more theoretical anslysis can potentially be done, rather than focusing the paper on empirical investigations.
3. The experimental section is unclear, making it difficult to identify the main takeaways. It reads more like a collection of observations than a focused evaluation of a central hypothesis. A large proportion of the section describes specific experiment set ups.

---

> ### Author Rebuttal · Authors · 2026-03-31
>
> We thank the reviewer for the constructive discussion. We address key concerns in the following sections:
>
> **Unsurprising for the theorist**
>
> While the high-level observation is widely held, our contribution is the formalization of the OOD train-test issue in SciML, the experimental validation methodology, and the fact that no amount of additional data helps.
> While these results may be unsurprising to a physics theorist, consider that a counterfactual result would have been unsurprising to an ML theorist. In the counterfactual, OOD error decreasing with data volume might have seemed "obvious" in hindsight due to intuition from Chinchilla scaling. Theorem 3.2 is in fact *anti*-intuitive in this sense: the convergence to a subspace projection holds *irrespective of dataset size*, contradicting the widely held expectation of ML practitioners that scaling data will help eventually. We believe transforming such folklore into precise, falsifiable statements with explicit dependencies is itself a contribution.
>
> **The results largely follow from standard tools... the contribution to the literature appears limited.**
>
> While the *tools* used are standard, the *results derived* are new.
> While the linear case does reduce to linear regression, we also address commonly used SciML models (MLPs, PINNs, DeepONets, FNOs, PI-DeepONets) which do not reduce to linear regression.
> Closed-form analysis is not readily tractable, the empirical investigation is not a substitute for theory; it is the necessary complement, providing evidence that the subspace-dependent failure mode persists in architectures where proofs are unavailable.
> To our knowledge, no prior work:
>
> 1. Analyzes the inverse problem from a gradient descent perspective with an eye for data "quality";
> 2. Proves that gradient descent converges to some projection for operator learning, making the subspace limitation completely independent of data quantity;
> 3. Shows that different model classes exhibit *opposing* generalization behaviors on the same problem (e.g., MLP diagonal vs. DeepONet lower-triangular vs. Finite Difference error increasing) via an extensive cross validation study (9 model classes trained across 25 datasets and other perturbations);
> 4. Provides interpretability via Green's function extraction (Figs. 9–10). We note that this can be used even on the most sophisticated of SciML models to determine whether the underlying physics has been learned.
>
> **Narrative arc / The experimental section is unclear... reads more like a collection of observations.**
>
> The paper follows the arc from the puzzle posed Figure 1 (why does a model with low training error learn the wrong operator) through developing a theoretical and experimental framework to land on a concrete explanation on how Green's functions extraction. The structure of the arc is as follows:
>
> 1. Figure 1 introduces the problem statement of not-quite-right convergence in a simple setting.
> 2. Theorems 3.1–3.2 establishes the behavior analytically and Section 4.1 replicates it empirically.
> 3. Sections 4.2 demonstrates the phenomenon persists in PINNs as parameter learning methods.
> 4. Section 4.3 demonstrates that the phenomenon persists in black-box operator learning methods, including physics informed architectures and losses.
> 5. Section Section 4.4 connects generalization to interpretability via Green's function extraction ( models that generalize yield interpretable operators), which fully closes the loop and directly explains the puzzle posed by the weight visualizations in Figure 1.
>
> We do acknowledge that the full loop of the arc was not explicit. We will clarify how each part in the experimental section contributes a piece to solving the puzzle.

---

> > ### Author Rebuttal · Reviewer_cBuA · 2026-04-02
> >
> > I appreciate the authors' clarifications regarding their contributions and will increase the Originality score accordingly.
> >
> > The authors claimed that the theoretical observations—which appear standard and nonsurprising—may be counter-intuitive for ML practitioners. As a theorist, I cannot definitively judge how widely these views are held or how impactful this specific theory is for the broader community. In light of this, I am lowering my Confidence score.
> >
> > I still believe the manuscript is not ready for publication. Since the goal is "transforming folklore into precise, falsifiable statements," the paper requires substantial restructuring to establish a coherent narrative; in its current form, the lack of a clear, central message remains a significant barrier to readability.

---

> > > ### Author Response · Authors · 2026-04-07
> > >
> > > We thank the reviewer for the thoughtful engagement and for reconsidering the originality assessment. We do agree that the narrative structure needs tightening, and we will restructure the paper to attempt to make the central arc much more explicit.

---

### Official Review · Reviewer_CHKG · 2026-03-13

**Soundness:** 3
**Presentation:** 3
**Significance:** 4
**Originality:** 3
**Overall Recommendation:** 5
**Confidence:** 4

**Summary:**

The paper proposes the interesting question of what the bounds of generalization are for physics machine learning models (e.g., Neural Operators and their variants). This work provides both theoretical and experimental results that demonstrate why the subspace spanned by the training data determines the quality of the learned model.

**Compliance With Llm Reviewing Policy:**

Affirmed.

**Key Questions For Authors:**

I think the paper is quite conclusive, therefore I only have some minor questions:
1. **Baseline choice:** the current set of baselines focuses on relatively simple linear PDEs. Why were nonlinear PDEs (e.g., Navier–Stokes) omitted from the manuscript?
2. **Unified perspective across methods:** do you have a more intuitive explanation for why this unified view can address generalization behavior for both PINNs and Neural Operators, for example? In my mind, these two represent very different types of SciML frameworks: one trains for the solution field directly, while the other trains a solver/operator. The nature of how they output a solution field is quite different to me.

**Limitations:**

There is no standalone limitations section, and the impact statement could use some improvements. For example, if we could better understand the generalization bounds in SciML, it could potentially lead to significant savings in computing resources.

**Strengths And Weaknesses:**

### Soundness
The authors introduce a unified view of PDE learning, $u = Af$, where $A$ can be a matrix, nonlinear mapping, or operator, which is clean and concise.

The theoretical analysis explaining how the learned operator effectively becomes a projection of the true operator onto the subspace spanned by the training data is insightful and well motivated. The empirical experiments also appear well designed, with evaluations across multiple model classes and function families.

One minor concern is that most of the results are relatively simple linear PDE settings.

### Presentation
I found the paper well written and easy to follow. In particular, the heatmap visualization is very intuitive to read and clearly illustrates the cross-subspace generalization behavior.

The only thing I think this paper could benefit from is a more unified view of SciML models. As it stands, the presentation seems to lean more toward a black-box (Neural Operator) view, while many SciML methods--such as PINNs, discretization-based solvers, and hybrid approaches--can be understood under a broader operator-learning or PDE-constrained optimization framework. Clarifying this connection could greatly strengthen the positioning of the work.

### Significance
This paper addresses one of the most important questions in SciML: the generalization ability of learned PDE solvers and operator models.

In practice, many physics ML systems perform well on test data drawn from similar distributions as the training data but fail when queried outside that distribution (OOD) (e.g., different IC, forcing, or parameters). By framing this issue through the lens of function-space subspaces and operator projections, the paper provides a useful perspective for understanding why such failures occur.

Even if the results are demonstrated mainly in simplified settings, the conceptual insight could influence how future SciML models are evaluated, particularly encouraging evaluation across different function families rather than relying only on in-distribution test sets.


### Originality
While it is common knowledge that most physics-driven models, whether data-driven or PDE-driven, often suffer when queried out of distribution (OOD), the key novelty lies not in proposing a new model but in providing a theoretical and experimental analysis connecting training-data subspaces, operator learning, and generalization behavior. The combination of operator-theoretic reasoning, cross-subspace experiments, and interpretability analysis provides a fresh and useful perspective on an existing but poorly understood phenomenon.

---

> ### Author Rebuttal · Authors · 2026-03-31
>
> We appreciate the reviewer's thorough and positive assessment. We address the two questions raised:
>
> **Q1: Why were nonlinear PDEs omitted?**
>
> Our positioning is that proof is required on linear problems before extending to nonlinear problems.
> The restriction to linear PDEs enables the rigorous theoretical analysis;
> the introduction of any nonlinearity does complicate the classical machinery used in PDE / numerical PDE analysis.
> We believe the presented proofs are contributions even with that contribution.
> Beyond the cross-validation methodology (training on one function class, evaluating on others) is directly applicable to nonlinear problems and constitutes a practical evaluation framework regardless of the underlying equation.
> We view the extension of the theory to mildly nonlinear settings (e.g., perturbative expansions around linear operators) as a natural and promising direction for future work.
> We note that the standard approach in numerical analysis is to establish convergence theory on linear ODEs and PDEs first before extending to more complex settings.
> Our paper follows this tradition.
>
> **Q2: Unified perspective across PINNs and Neural Operators.**
>
> We agree with the reviewer's perspective on the difference between PINNs and solution operator methods.
> Our framing is to put the PINN in the same category finite difference class of models: operationally, the PINN outputs a solution field similarly to how a finite difference method has to solve the equation.
> Our results show that the PINN and finite difference model have similar generalization patterns that a very distinct from the Neural Operators and black-box models, which gives a concrete ramification of this crucial difference.
>
> **Limitations and impact.**
>
> We agree that a standalone limitations section would improve the paper and we thank the reviewer for the suggestions. We will revise these sections as suggested.

---

> > ### Author Rebuttal · Reviewer_CHKG · 2026-04-04
> >
> > I appreciate the authors’ clarification.
> >
> > I am afraid I still don't follow why PINN and finite difference are under the same class, other than that they are "white-box." Can you elaborate more on `operationally, the PINN outputs a solution field similarly to how a finite difference method has to solve the equation`?
> >
> > Also, another way to put my Q2 **Unified perspective across methods** is, if the focus of this paper is to understand black-box models (as everything before Sec. 4 suggests), then why is PINN included in this study in the first place? My understanding is that, in theory, you could rewrite some "white-box" models as "black-box" (Eqn. 4) as an operator (e.g., a finite difference stencil can be rewritten as a linear operator). I am not sure what the role of PINN is here.

---

> > > ### Author Response · Authors · 2026-04-07
> > >
> > > We thank the reviewer for the follow-up and will try to clarify.
> > >
> > > **PINNs and FDM**: we grouped PINNs and traditional numerical methods moreso because the philosophy behind PINNs is the same with FDM/FEM (finite difference/finite element method): given a specific problem on a specific domain, return only the solution vector. FDM/FEM utilizes the compute on linear solves, while PINNs perform gradient descent.
> > >
> > > If you allow us to entertain FEM instead for a second, there is a slightly deeper connection. Both can be formulated as a minimization problem, where FEM is over (usually) piecewise polys while PINNs is just a minimization over the trial space of MLPs. In fact, the whole least square FEM field is quite similar in taste to PINNs, but taking theory from there and applying it to PINNs is difficult as the space of neural networks is not even vector spaces, detrimental to many proofs! (e.g. see middle of page 3 of 10.1137/20M1366587).
> > >
> > > PINNs/FDM/FEM contrasted against the other operator learning approaches, where once the method is trained, replacing the right-hand side with any other function is almost trivial. Perhaps one way to tie DeepONet and FNO is with reduce-order modelling (ROM), where rather than learning the physics directly, we learn a representation which is fast to inference.
> > >
> > > **On why PINNs are included**: mostly due to the ubiquitousness of PINNs in current literature. We do want to point out that PINNs actually cannot be put in the form of Eqn 4) for the standard MLPs,
> > > i) a trained PINNs on a single f is useless on a different forcing f’ ii) there is no “basis” for the space of MLPs (as it’s not even a vector space). Thus, while it’s more “white-box”, than other methods, it is really the nonlinearities that really put it slightly different on the spectrum compared to traditional methods.

---

### Decision · Program_Chairs · 2026-04-30

**Decision:**

Accept (regular)

**Comment:**

The paper offers a useful theoretical and empirical perspective on generalization in scientific machine learning, highlighting the critical role of training-data function spaces and connecting this to interpretability.

Although some reviewers found the presentation and narrative in need of improvement, the core contribution was considered meaningful and the rebuttal clarified the scope and significance of the work.